# Single-Cell Analysis Reveals that Chronic Silver Nanoparticle Exposure Induces Cell Division Defects in Human Epithelial Cells

**DOI:** 10.3390/ijerph16112061

**Published:** 2019-06-11

**Authors:** Ellen B. Garcia, Cynthia Alms, Albert W. Hinman, Conor Kelly, Adam Smith, Marina Vance, Jadranka Loncarek, Linsey C. Marr, Daniela Cimini

**Affiliations:** 1Department of Biological Sciences and Fralin Life Sciences Institute, Virginia Tech, Blacksburg, VA 24061, USA; ebgarcia@vt.edu (E.B.G.); calms1@umbc.edu (C.A.); ahinman@stanford.edu (A.W.H.); conork@vt.edu (C.K.); adam747@vt.edu (A.S.); 2Department of Civil and Environmental Engineering, Virginia Tech, Blacksburg, VA 24061, USA; Marina.Vance@colorado.edu (M.V.); lmarr@vt.edu (L.C.M.); 3Center for Cancer Research, National Institute of Health, Frederick, MD 21702, USA; jadranka.loncarek@nih.gov

**Keywords:** silver nanoparticles, mitosis, nanotoxicology, micronucleus, tetraploidy

## Abstract

Multiple organizations have urged a paradigm shift from traditional, whole animal, chemical safety testing to alternative methods. Although these forward-looking methods exist for risk assessment and predication, animal testing is still the preferred method and will remain so until more robust cellular and computational methods are established. To meet this need, we aimed to develop a new, cell division-focused approach based on the idea that defective cell division may be a better predictor of risk than traditional measurements. To develop such an approach, we investigated the toxicity of silver nanoparticles (AgNPs) on human epithelial cells. AgNPs are the type of nanoparticle most widely employed in consumer and medical products, yet toxicity reports are still confounding. Cells were exposed to a range of AgNP doses for both short- and-long term exposure times. The analysis of treated cell populations identified an effect on cell division and the emergence of abnormal nuclear morphologies, including micronuclei and binucleated cells. Overall, our results indicate that AgNPs impair cell division, not only further confirming toxicity to human cells, but also highlighting the propagation of adverse phenotypes within the cell population. Furthermore, this work illustrates that cell division-based analysis will be an important addition to future toxicology studies.

## 1. Introduction

As the number of manmade chemicals and materials is ever increasing, massive amounts of health risk assessments are needed to ensure safety to human populations, other organisms, and our environment [1]. The majority of toxicity data, past and present, come from whole animal models [2,3,4,5]. However, the amount of information required by law and demanded from the public about chemical safety has become too large of an undertaking for animal models. Moreover, animal models pose serious ethical issues and are expensive, time consuming, and scientifically questionable [6]. Animals have proved valuable for the last 80 years, but toxicology is at a pivotal point in history. Given the concerns surrounding animal testing, the toxicology community is currently committed to developing novel, non-animal tests, with a particular focus on cellular and computational methods [3,6,7,8,9,10].

Many in vitro toxicology approaches have been used in the past [11,12], but they are not without limitations, being frequently based on endpoints that are highly specific and/or cannot reveal the underlying mechanisms of cellular toxicity. For instance, many cellular toxicology studies employ assays that measure some form of cell death, typically apoptosis [11,12,13,14]. These assays are important because excessive cell death will disrupt tissue homeostasis. However, such assays would not reveal damage that impairs cell function without causing cell death. Other assays (e.g., comet assay) measure DNA damage [11,14,15,16], which is an important metric because, if not repaired, DNA damage can affect cell physiology in many ways, and can potentially lead to cell transformation and tumorigenesis. Some studies measure perturbations of the cell cycle, typically by FACS analysis [11,14,17]. This type of analysis is informative, because it can hint at the specific cellular processes that are being affected. However, if not followed up by further investigations, cell cycle analysis alone will not provide information about specific defects or cell-to-cell variability. Another commonly used assay is the micronucleus assay [11,12,18,19], which measures genetic damage that can potentially affect subsequent cell generations [20]. However, if not combined with a method aimed at discriminating the origin and nature of the micronuclei, this assay cannot provide insight into the underlying cause of micronucleus formation. Concomitant immunostaining for kinetochore proteins is one strategy that has been previously employed to gain insight into the nature of micronuclei [21,22]. Other than the micronucleus test, there are no other assays that are commonly used for cellular toxicology that provide, as a readout, a measure of genetic damage that has arisen during cell division and that is inherited by the daughter cells. However, inherited genetic damage may be a better readout of the impact of exposure than other commonly used readouts, such as cell death. Indeed, whereas cell death would simply eliminate cells from a population, accumulation of genetic damage and genome instability in a cell population may represent the first step towards the emergence of diseases, such as cancer [23]. For this reason, we aimed to develop a cell division-focused approach and did so while investigating the effects of silver nanoparticle (AgNP) exposure on human cells.

AgNPs are used in a number of consumer and medical products due to their desirable anti-bacterial and anti-fungal properties [24,25,26,27,28,29,30,31], but may be taken up into the body via inhalation, ingestion, or dermal absorption [26,32,33]. Regardless of the exposure route, the hazard of AgNP exposure substantially increases if the nanoparticles are delivered to the interior of the cell. The most common process for AgNP cellular internalization is believed to be the Trojan horse mechanism [34,35,36,37]. Particles become coated with macromolecules, such as proteins or growth factors, found in biological serum and the cell recognizes and engulfs these macromolecules, unknowingly also engulfing AgNPs. Numerous in vitro studies have shown that AgNPs can induce both cytotoxicity and genotoxicity in a variety of mammalian cells [25,32,37,38,39,40,41,42,43,44,45], as well as effects on differentiation state [46]. Specifically, there are reports of AgNP-dependent cell cycle arrest in G1 [47,48], S-phase [49,50], and at the G2/M transition [39,51]. Some studies have shown AgNP exposure to cause micronuclei and chromosome aberrations, which the authors attributed to DNA damage [39,52,53]. Other groups, however, found no effects of AgNP exposure on some cell types, such as human astrocytes [54] and human keratinocytes [55].

Despite the large amount of literature describing AgNP toxicity, the effects of AgNPs on cell division have not been investigated before. Therefore, we employed a cell division-focused, single-cell approach to investigate the effects of AgNP exposure on RPE-1 cells, a human, non-transformed, hTERT-immortalized, retinal pigmented epithelial cell line that is considered the gold standard to study cell division in non-transformed cells. It is important to use such a standard experimental model of cell division as a first step toward optimizing cell division toxicology approaches, with the goal to later apply these approaches to other experimental models, including primary cell lines and whole tissue models. We conceived an experimental design that consisted of both short- and long-term exposure with a range of AgNP doses. To evaluate the effects of this exposure, we used a combination of approaches, which included cell population analysis of cell death and proliferation, as well as microscopy-based single-cell analysis of both live and fixed cells aimed at identifying errors arising during cell division [56].

## 2. Materials and Methods

### 2.1. Silver Nanoparticle Characterization

Silver nanopowder, composed of 99% silver and 0.3% Polyvinylpyrrolidone (PVP), was obtained from Nanostructured and Amorphous Materials, Inc. (#0478HW, TX, USA). The silver nanopowder was characterized by the manufacturer, and was specified, based on TEM analysis, to contain 20 nm sized AgNPs. To make a liquid suspension of AgNPs, 2 mg of silver nanopowder was added to 1 mL of sterile-filtered deionized water and sonicated for approximately 20 min. Silver nanopowder suspensions were characterized for the hydrodiameter (nm), polydispersity index (PDI), and zeta potential (mV) using dynamic light scattering (DLS; Zetasizer Nano Series, Malvern Instruments Ltd., Worcestershire, UK). Additionally, we characterized AgNPs suspended in cell culture medium. To do this, 2 mg of silver nanopowder was added to 1 mL of sterile DMEM: F12 cell culture medium and sonicated for approximately 20 min. This suspension was then characterized using DLS. Lastly, we took a 2 mg/mL stock of AgNPs suspended in water, diluted it in sterile water 1:1, as well as cell culture medium to 15, 25, and 50 µg/mL and performed measurements. Each DLS measurement was repeated three times by the instrument and then averaged. A total of 12 different suspensions from four different batches of silver nanopowder were measured. The data reported represent the mean ± SEM of the twelve independent DLS measurements. The silver nanopowder came with a six-month expiration date; therefore, fresh nanopowder was routinely purchased and particle sizes were validated using DLS (not all data shown).

### 2.2. Cell Culture and Experimental Design

Human hTERT-immortalized retinal pigmented epithelial (RPE-1) cells were obtained from American Type Culture Collection (ATCC, BA, USA). RPE-1 cells were maintained in DMEM:F-12 medium, supplemented with 10% fetal bovine serum and 1X antibiotic-antimycotic solution (all from Gibco, Life Technologies, CA, USA) in a humidified incubator at 37 C and with 5% CO_2_. RPE-1 cells were seeded for all experiments at approximately 500,000 cells when grown in T-25 cell culture flasks (Corning Life Sciences, NY, USA) or 250,000 cells when grown in 35 mm six-well plates (Corning Life Sciences, NY, USA), 35 mm glass-bottom Petri dishes (MatTek, MA, USA), or on glass coverslips inside 35 mm Petri dishes (Corning Life Sciences, NY, USA). Twenty-four hours following seeding, cells were treated with a range of AgNP doses (0.5, 1, 2, 5, 10, 15, 25, 50, 75, and 100 µg/mL). The AgNP doses were selected to mirror concentrations from other in vitro AgNP toxicology studies, as well as explore lower, potentially more realistic doses (24–26,29–32,37–45). Control cells were not treated with AgNPs, but the medium was replaced with fresh medium at the time of treatment. Three experimental designs were conceived for this study and referred to as acute, moderate, and chronic AgNP exposure (Figure 1) and the cells were treated with a range of AgNP doses during each of these three exposure schemes. For acute AgNP exposure, cells were cultured in the presence of AgNP for one 24 h period (Figure 1a). This experiment was intended to look for any immediate signs of toxicity from AgNPs. For moderate AgNP exposure, cells were administered six 24 h AgNP treatments at regular intervals over a period of three weeks (Figure 1b). For chronic AgNP exposure, cells were administered twelve 24 h AgNP treatments at regular intervals over a period of six weeks (Figure 1b). These longer exposures were aimed at unveiling any effect that may accumulate over time. To investigate the effects of acute AgNP exposure on cell division, live-cell imaging was performed during the 24 h treatment and during the 24 h following AgNP washout (Figure 1a). Washouts consisted of three washes in PBS (Phosphate Buffer Saline: 140 mM NaCl, 2.5 mM KCl, 15 mM Na_2_HPO_4_ 7H_2_O, 1.6 mM KH_2_PO_4_, pH 7.2). Cells were also collected for fixed-cell analysis at the end of the 24 h treatment (Figure 1a). Additionally, we performed mass spectrometry analysis on the cell culture at the end of the treatment and 24 h after washout (Figure 1a). For moderate exposure, data collection (excluding mass spectrometry) was conducted in the same fashion as described for acute exposure (Figure 1a), except that the cells being analyzed were undergoing or had completed a sixth AgNP treatment. For chronic exposure, data collection (excluding mass spectrometry) was conducted in the same fashion as described for the acute exposure (Figure 1a), except that the cells being analyzed where undergoing or had completed a twelfth AgNP treatment.

### 2.3. ICP-MS 

RPE-1 cells were grown in sterile, T-25 cell culture flasks (Corning Life Sciences, NY, USA). ICP-MS analysis was performed both on the adherent cells and on the medium in which the cells were grown, which we expected to contain dead, floating cells. For analysis of floating cells, both the medium and the PBS used for two washes (totaling 15 mL) were collected into individual conical tubes. For analysis of the adherent cells, cells were trypsinized, resuspended in PBS, and collected into separate conical tubes. All tubes were centrifuged at 1,000 rpm for 5 min. The supernatant was aspirated and the pellet was resuspended in 2 mL of aqua regia (1:3, 15 M HNO_3_:12 M HCl) to dissolve any residual AgNPs. From this suspension, 50 µL was further diluted into deionized water, to a final volume of 5 mL. Samples were analyzed for metal (specifically silver) concentrations using a Thermo Electron X-Series inductively coupled plasma mass spectrometer (ICP-MS) per standard method 3125-B (APHA, AWWA, and WEF, 1998). Samples and calibration standards were prepared in a matrix of 2% nitric acid by volume. All chemical reagents used were purchased from Fisher Scientific (NH, USA). Two technical replicates of ICP-MS measurements were averaged for each individual sample from each experiment and the data are reported as mean ± SEM from two independent experiments.

### 2.4. Transmission Electron Microscopy 

RPE-1 cells were grown in sterile, 35 mm six-well plates. Control and treated cells were first washed with PBS three times, then fixed directly in six-well dishes using Karnovsky fixative (16% formaldehyde, 50% glutaraldehyde, 0.2 M cacodylate buffer, pH 7.4) and stored at 4 °C until further processing. Post fixation, embedding and serial sectioning were performed according to established protocols. Sections (90 nm) were cut and examined using a Hitachi transmission electron microscope operating at 80 kV. 

### 2.5. Growth Curves

To determine the proliferative capacity of RPE-1 cells during and after acute treatment or the last treatment of either moderate or chronic exposure, cells were plated in six-well plates at a density of 40,000 cells per well. Cells were counted a first time 12 h after initial seeding (indicated as day 0) and every 24 h thereafter for up to five days. The 24 h AgNP treatment was performed at day two (i.e., two days after the initial counting) and washout was performed at day three. For counting, the cells were washed twice with PBS and trypsinized. Cells were then re-suspended in medium to a final total volume of 1 mL. From this, a 100 µL sample was thoroughly pipetted and mixed with an equal amount of 0.4% trypan blue solution (Gibco, NY, USA), and subjected to cell counting. Cell counts were performed using a hemocytometer (Bright-Line, PA, USA). Counts were obtained from two technical replicates for each sample and data are reported as the mean ± SEM from two independent experiments.

### 2.6. Live Cell Imaging and Analysis

RPE-1 cells were grown on sterile, 35 mm glass-bottom Petri dishes. Twenty-four hours following cell passage, untreated cells were imaged for 24 h in fresh medium, whereas treated cells were imaged during the 24 h AgNP treatment, as well as during the 24 h following AgNP washout (recovery). Imaging was performed in L-15 medium (Gibco, Life Technologies, CA, USA) supplemented with 4.5 g/L glucose at 37 °C using an enclosed humidified stage top incubator (Tokai Hit, model #INUBG2ATFP-WSKW, Japan). Images were acquired using a Nikon Eclipse Ti inverted microscope (Nikon Instruments Inc., NY, USA) equipped with the following: (i) phase-contrast transillumination, (ii) Lambda Smart shutter controller (Sutter Instruments, CA, USA), (iii) Uniblitz shutter driver, (iv) automated ProScan stage (Prior Scientific, Cambridge, UK), (v) 20X/0.3 NA A-Plan corrected phase contrast objective (Nikon Instruments Inc., NY, USA), and (vi) CoolSNAP-HQ2 CCD camera (Photometrics, AZ, USA). Six-to-ten different fields of view were imaged by phase-contrast every four min for approximately 24 h. 

Time-lapse videos were analyzed for mitotic timing and behavior using NIS Elements AR software (Nikon Instruments Inc., NY, USA) on a PC computer. Mitotic timing was quantified as the time elapsed between the beginning of cell rounding and beginning of cleavage furrow ingression. Videos were also surveyed for any distinct cell behaviors. The behaviors quantified in all live cell experiments were the following: normal mitosis, mitotic arrest (cell rounding lasting longer than 3 h), and cell death. Live-cell imaging was performed for two independent experiments and a total of 200 cells from different fields were analyzed in each experiment. Cell death and mitotic arrest data are reported as the mean – SEM and the mitotic timing data are reported as mean ± SEM from the two independent experiments.

### 2.7. Trypan Blue Assay for Cell Viability

As a control for potential effects from silver ions, cells were treated with 50 µg/mL of AgNO_3_ for 24 h with or without a silver chelator (N-Acetyl-L-cysteine, NAC; Sigma Aldrich, MO, USA). To make a stock solution of AgNO_3_, 0.5 g of AgNO_3_ (Fisher Scientific, NH, USA) was dissolved in 150 mL of sterile deionized water and sterile filtered. The stock solution was then further diluted in cell culture medium to a working dilution of 50 µg/mL. To make a stock solution of NAC, 100 mg of powder NAC was dissolved in 1 mL of water, heated, and then sterile filtered. The stock solution was then diluted in cell culture medium to a working solution of 10 mM.

Following acute treatment with AgNPs (+/- NAC) or AgNO_3_ (+/- NAC), RPE-1 cells were gently washed with PBS two times to collect any non-adherent cells, not including mitotic cells. The original medium and washes were combined and the solution was spun down at 1000 rpm for 5 min. The supernatant was aspirated and the cell pellet was suspended in 1 mL of fresh medium. Cells were trypsinized, the cell suspension was centrifuged at 1000 rpm for 5 min, and the cell pellet was suspended in 1 mL of fresh medium. A 100 µL aliquot of either cell suspension was diluted 1:1 in 0.4% trypan blue (Sigma-Aldrich, MO, USA), which stains dead cells blue, while live cells remain translucent. Cell counts were performed by hemocytometer (Bright-Line, PA, USA) and averaged from two technical replicates per sample and data are reported as the mean - SEM from two independent experiments.

### 2.8. Immunostaining, Fluorescence Microscopy, and Fixed-Cell Analysis

RPE-1 cells were grown on sterile glass coverslips inside 35 mm Petri dishes. For immunostaining, cells were fixed using freshly prepared 4% paraformaldehyde dissolved in PHEM (60mM PIPES, 25 mM HEPES, 10 mM EGTA, 2 mM MgSO4, pH 7.0), then permeabilized in PHEM containing 0.5% Triton-X, and blocked in 10% boiled goat serum (BGS) in PHEM for 1 h at room temperature. Following blocking, cells were immediately transferred into a humidified chamber and incubated with primary antibodies overnight at 4 °C. After primary antibody incubation, cells were washed three times in PBST (PBS + 0.05% tween 20) and then incubated with secondary antibodies. Finally, the coverslips were counterstained with DAPI (4’,6-Diamidino-2-Phenylindole, Dihydrochloride; 5 ng/mL) for 10 min, rinsed, and mounted onto microscope slides (Fisher Scientific, NH, USA) in an antifade solution containing 90% glycerol and 0.5% N-propyl gallate. The primary antibodies used were human anti-centromere (Antibodies Inc., CA, USA) at a 1:100 dilution and mouse anti-α tubulin (DM1A, Sigma Aldrich, MO, USA) at a 1:500 dilution. The secondary antibodies used were RhodamineX goat-anti-human (Jackson ImmunoResearch Laboratories, Inc., PA, USA) at a 1:200 dilution and Alexa488 goat-anti-mouse (Molecular Probes, Life Technologies, CA, USA) at a 1:400 dilution. All antibodies were diluted in 5% BGS in PHEM.

Immunofluorescently labeled RPE-1 cells were viewed using a Nikon Eclipse TE2000-U inverted microscope (Nikon Instruments Inc., NY, USA) equipped with a 60x/1.4 NA Plan-A objective and an automated ProScan stage (Prior Scientific, Cambridge, UK). Illumination was achieved using an X-cite 120Q light source (Excelitas Technologies, MA, USA.). Immunofluorescently labeled cells were imaged using the same microscope through a swept field confocal system (Prairie Technologies, WI, USA). The confocal system was accessorized with multiband pass filter set for illumination at 405, 488, 561, and 640 nm, and illumination was obtained through an Agilent (CA, USA) monolithic laser combiner (MLC400) controlled by a four channel acoustic-optic tunable filter. Images were acquired using a HQ2 CCD camera (Photometrics, AZ, USA). Fixed-cell experiments were repeated in duplicate and approximately 1000 interphase cells were analyzed blindly in each experiment. MNi and nuclear phenotype data are both reported as the mean + SEM.

## 3. Results

The AgNPs used in this study were obtained from a commercial source and were characterized by the manufacturer (Nanostructured and Amorphous Material, Inc.), but we performed some additional characterization. Dynamic light scattering measurements of AgNPs suspended in water indicated an average particle size of 70.15 nm (± 21.81 nm), with a size range of 44–95 nm (Appendix A). This size is larger than the 20 nm diameter measured by the manufacturer, which indicates some degree of aggregation for AgNPs in aqueous suspension. However, there was no significant variation in the measurements taken from the same batch at different time points or from different batches (Appendix A). The dispersity index of the suspensions met quality criteria (< 0.7 PDI) and the zeta potential was negative, as expected for AgNPs (Appendix A). Particles suspended at high concentrations in cell culture medium showed larger diameter than observed for particles suspended in water (Appendix A). This is consistent with reports that AgNP aggregation occurs in cell culture media [57,58]. However, when the stock suspension was diluted in media at concentrations that were within our experimental range, the diameter measured was substantially lower than that observed in the stock suspension and closer to the diameter reported by the manufacturer (Appendix A), suggesting that minimal aggregation occurred at the concentrations used in our experiments. For the suspensions in tissue culture medium, PDI met quality criteria and the zeta potential reflected a negative charge (Appendix A).

### 3.1. Silver Nanoparticle Internalization and Accumulation

Particle uptake by the cells could be inferred from time-lapse videos of live cells imaged by phase contrast microscopy. Indeed, highly refractive dots could be detected in and around cells that had been treated with AgNPs, but not in untreated cells (Figure 2a); we inferred that these dots corresponded to AgNPs. Furthermore, analysis of time-lapse videos (Figure 2a,b and Appendix A) showed that 100% of cells in acutely treated samples engulfed AgNPs (Figure 2b and Appendix A). Further evidence of particle internalization was obtained from cells with fluorescently stained DNA and microtubules showing dark fluorescence-exclusion regions (Figure 2c, middle and right panels), which likely represented internalized AgNPs. These dark regions were not observed in untreated cells (Figure 2c, left panel), but were evident in both interphase (Figure 2c, middle panel) and mitotic (Figure 2c, right panel) AgNP-treated cells. Finally, confirmation that the AgNPs were intracellular (as opposed to outside the cell or adhered to the cell surface) was provided by transmission electron microscopy, which showed the presence of phase-dark material enclosed in vesicles randomly distributed in the cytoplasm (Figure 2d).

To obtain a more quantitative measure of AgNPs accumulated by the cells upon treatment, we used inductively coupled plasma mass spectrometry (ICP-MS) to quantify the total concentration of silver in either adherent cells or the culture medium (which would include floating, non-adherent cells). This analysis was performed both at the end of one 24 h treatment (acute exposure) and after recovery. As expected, we found that as the dose of AgNPs increased, the total amount of silver (in adherent cells, plus medium and washes) increased proportionally (Figure 3a, combined dark and light bars). Importantly, at the highest dose (75 µg/mL), the total amount of silver in the cell culture equaled the amount of silver in a suspension of AgNPs alone prepared at the same concentration. However, the concentration of silver in adherent cells decreased at the highest dose after increasing at the two lower doses (Figure 3a, light bars), whereas a large amount of silver was present in the medium/washes fraction (Figure 3a, 75 μg/mL, dark bar). This indicates that AgNPs at a dose of 75 µg/mL can induce high rates of cell death, which was subsequently confirmed in additional experiments (see next results section). When this analysis was performed 24 h after washout, the amount of silver in the adherent cells was only slightly lower than that observed at the end of the 24 hr treatment (compare Figure 3a,b, light bars), suggesting that once cells took up AgNPs, they did not expel it. A small amount of silver was detected in the medium 24 h after washout (Figure 3b, dark bars), indicating low rates of cell death still occurring during recovery. When the amount of intracellular silver was normalized to the number of adherent cells, a clear dose-dependent increase in the amount of silver per cell was apparent (Figure 3c, dark bars). However, the amount of silver per cell 24 h after washout was 50% or less than the amount of silver per cell at the end of a 24 h treatment (Figure 3c, light bars). This can be explained by the fact that during recovery, cells keep proliferating and therefore the total amount of intracellular silver will be distributed among a larger number of cells.

Overall, within the timeframe and the dose range of our experiments, these data indicate the following: (i) RPE-1 cells uptake AgNPs from the surrounding environment; (ii) the amount of silver taken up by individual RPE-1 cells is proportional to the amount of AgNPs available; (iii) once RPE-1 cells uptake the AgNPs, they will not expel them.

### 3.2. Reduced Cell Viability and Mitotic Delay from AgNP Exposure

To assess the effect of AgNP exposure on cell proliferation, we evaluated the proliferative capacity of untreated cells compared to cells treated with AgNPs at increasing doses and over different exposure durations (Figure 4, Figure 5 and Figure 6). Control cells were highly proliferative; in contrast, acute exposure to AgNPs induced a concentration-dependent decrease in cell proliferation (Figure 4a). Noticeably, cells treated with the highest dose of AgNPs (75 μg/mL) displayed a negative proliferation rate during the exposure time window; however, this was not observed in cells treated with lower doses. Moreover, even cells treated with the highest dose regained a positive proliferation rate upon AgNP washout. A dose of 100 μg/mL of AgNPs was also tested, but resulted in a complete lack of cell survival.

To further investigate the effect of acute AgNP exposure on cell proliferation, we performed time-lapse imaging experiments both during the 24 h treatment and during the 24 h following washout. Cells treated with AgNPs displayed a variety of cell behaviors, including normal mitoses (Figure 4b), mitotic arrest (Figure 4c), and a phenotype indicative of cell death (Figure 4d). The latter two behaviors were quantified as the fractions of cells that rounded up and remained rounded for more than 3 h (mitotic arrest) or rounded up and then died during the 24 h period (cell death) among all the cells entering mitosis. As the dose of AgNPs increased, an increasing number of cells either became arrested or died during acute AgNP exposure (Figure 4e, dark stacked bars). While mitotic cells were observed over the entire 24 h of imaging, adverse cellular effects began 3 h or more into the AgNP treatment (Appendix A). During recovery from acute AgNP exposure, cells were still observed to arrest in mitosis and die (Figure 4e, light stacked bars), despite the overall resumption of proliferation.

Finally, we measured mitotic timing by determining the elapsed time between cell round up (mitotic entry) and anaphase onset in cells completing mitosis during the 24 h period of imaging. Cells acutely treated with AgNPs displayed a modest, but significant concentration-dependent increase in mitotic timing (Figure 4f, dark bars). Specifically, mitotic timing was 26.82 ± 0.47 min (mean ± SEM) for control cells and increased for treated cells to 27.73 ± 0.47, 30.53 ± 0.78, 31.18 ± 0.63, and 31.60 ± 0.55 min, respective to increasing AgNP dose. Mitotic timing comparable to that of control cells was re-established after AgNP washout (Figure 4f, light bars). Overall, these results indicate that although mitotic timing is quickly restored after a single 24 h AgNP treatment, some adverse effects (i.e., mitotic arrest and death) on the cell population persist even after AgNP washout. This latter observation could partly explain the overall effects on cell proliferation.

To exclude the possibility that the adverse effects observed in our experiments may be due to the release of silver ions from the AgNPs instead of the NPs themselves, we tested the effects of AgNO_3_ (a positive control for silver ions) and AgNPs with or without the silver ion chelator NAC [59] on cell viability via a trypan blue assay (see methods for details). A dose of 50 μg/mL AgNO_3_ resulted in a complete lack of cell survival, but cell viability was not affected when AgNO_3_ was combined with NAC (Appendix A), indicating that NAC effectively chelated silver ions. We next compared cell viability in adherent (Appendix A) and non-adherent (Appendix A) cells treated with AgNPs alone (Appendix A) or with AgNPs and NAC (Appendix A). Cell death was detected at very low levels in the adherent cells and those levels were not significantly different (χ^2^, *p* > 0.05 for all pairwise comparisons) in cells co-treated with NAC (Appendix A). The non-adherent cells treated with AgNPs displayed high levels of cell death that were dose-dependent (Appendix A). However, the fraction of dead cells was not significantly different (χ^2^, *p* > 0.05 for all pairwise comparisons) in cells co-treated with NAC (Appendix A) compared to cells treated with AgNPs alone (Appendix A). These observations led us to conclude that the effects from AgNP treatment were due to the NPs themselves and not to silver ions released by the AgNPs.

Growth curves, analysis of mitotic behavior, and mitotic timing were also obtained for moderately exposed cells undergoing or following a sixth AgNP treatment. Similar to what we observed for acute exposure, cells moderately exposed to 75 μg/mL of AgNPs experienced a negative proliferation rate during treatment (Figure 5a). However, in moderately exposed cells, even lower concentrations of AgNPs (15 and 25 μg/mL) induced a decrease in the proliferation rate during treatment (Figure 5a), which was not the case for acutely exposed cells at these lower concentrations (see Figure 4a). At all doses, proliferation resumed after AgNP washout (Figure 5a). These moderately exposed cells also displayed mitotic arrest or death in a dose-dependent manner during treatment (Figure 5b). However, the cells appeared to recover more quickly compared to acutely treated cells (compare Figure 5b and Figure 4e), possibly suggesting some form of “adaptation” to the adverse effects of AgNPs. For moderate exposure, a statistically significant increase in mitotic timing was observed only in cells treated with 75 μg/mL of AgNPs (Figure 5c), for which mitotic timing was 30.60 ± 0.71 min (mean ± SEM) compared to 25.40 ± 0.53 min in control cells (Figure 5c). Cells regained normal mitotic timing during recovery from AgNP exposure, although a slight elevation was observed in cells recovering from a 25 μg/mL AgNP moderate treatment (Figure 5c). In summary, these results indicate that moderate exposure to AgNPs impacts the dynamics of cell proliferation (Figure 5a). A general resumption of proliferation capacity is observed during recovery, although observations at the single-cell level indicate that mitotic arrest and cell death continue to occur even after AgNP washout (Figure 5b), but at lower rates than we observed for acutely treated cells.

Growth curves, analysis of mitotic behavior, and mitotic timing were similarly obtained for chronically exposed cells undergoing or following a twelfth AgNP treatment. Overall, the growth curves (Figure 6a) were similar to those obtained upon acute exposure, where the highest dose of treatment led to a negative proliferation rate during treatment and a slower regain of positive proliferation during recovery (compare Figure 6a to Figure 4a). There was a dose-dependent increase in the fraction of cells that experienced mitotic arrest or cell death (Figure 6b, dark stacked bars) and the mitotic arrest and cell death phenotypes persisted after AgNP washout (Figure 6b, light stacked bars), particularly at the highest dose. The most dramatic effect observed in these chronically exposed cells was a very significant increase in the mitotic timing (70.76 ± 3.69 min) in cells treated with the highest AgNP dose (Figure 6c, dark bars). The mitotic timing in these cells was over twice that of control cells (25.00 ± 0.30 min) or that of acutely (31.60 ± 0.55 min) and moderately exposed (30.60 ± 0.71 min) cells at the same dose. Nevertheless, mitotic timing returned to normal upon AgNP washout (Figure 6c, light bars). Overall, these results indicate that chronic exposure to AgNPs continues to impact the dynamics of cell proliferation and the magnitude of this effect is concentration-dependent.

### 3.3. Emergence of Abnormal Nuclear Phenotypes in AgNP-Treated Cell Populations

In parallel to the analysis of proliferation, mitotic behavior, and timing, we prepared fixed samples from cells treated with a larger range of AgNP doses and performed an analysis aimed at identifying nuclear phenotypes that may emerge from defective cell division [56]. Specifically, we quantified the frequency of micronuclei (Figure 7a), which may result from mitotic errors occurring at the single chromosome level, and the frequency of abnormal nuclei (Figure 8a,b) that may arise from large-scale cell division defects, such as mitotic slippage (mitotic exit without chromosome segregation) or cytokinesis/abscission failure. Micronucleus analysis (Figure 7b) showed that micronuclei are very rare in untreated cells. However, the fraction of micronucleated cells was significantly higher in many of the treated samples (Figure 7b). For acutely treated cells, an increase in micronucleated cells was observed, but, in most cases, the increase was not statistically significant compared to the untreated cells. However, in cells exposed to moderate and chronic treatments, the number of micronucleated cells was substantially increased even at low doses. Indeed, in chronically exposed cells, even a dose as low as 0.5 μg/mL induced a 4.5-fold increase in the number of cells with micronuclei. Surprisingly, we did not observe a dose-dependent relation (i.e., no statistically significant differences were observed between doses within individual treatment regimens). To discriminate between micronuclei originating from chromosome missegregation and micronuclei originating from DNA damage [60], we immunostained for kinetochore proteins and quantified kinetochore-positive (MN+, Figure 7a, top) vs. kinetochore-negative (MN-, Figure 7a, bottom) micronuclei. This analysis showed that the fraction of cells with kinetochore-positive micronuclei (Figure 7b, dark bars) was generally greater than the fraction of cells with kinetochore-negative micronuclei (Figure 7b, light bars), indicating that micronucleus formation due to missegregation of whole chromosomes was a more common event than micronucleus formation due to the presence of chromosome fragments.

To identify the possible occurrence of large-scale cell division defects, we analyzed interphase nuclei and identified abnormal phenotypes. The common abnormal nuclear phenotypes that we observed included binucleate, multinucleate, pinched, and lobed nuclei (Figure 8a). In many cases, the total amount of DNA in these cells with abnormally shaped nuclei appeared to be larger than (about twice) the amount of DNA in cells with single, normally shaped nuclei. The percentage of cells with abnormally shaped nuclei was significantly higher (up to ~five-fold) in AgNP-treated cells compared to the controls (Figure 8b). Similar to what we found for micronuclei, even at very low AgNP concentrations but longer exposure times, a substantial increase in the number of cells with these abnormal nuclear phenotypes could be observed (Figure 8b, green bars). Noticeably, we were able to gather evidence that some of these cells with abnormal nuclei could re-enter mitosis (Figure 8c and Appendix A), indicating that exposure to AgNPs, even at very low doses, could have long-lasting effects on the cell population if exposure occurs for a prolonged period of time.

## 4. Discussion

### 4.1. Long-Term in vitro Exposure to AgNPs, Even at Very Low Doses, Leads to the Accumulation of Cells with Abnormal DNA Content

Long-term exposure to AgNPs has been investigated in a number of in vivo studies, which found prolonged exposure to AgNPs to cause changes in expression of genes associated with neurodegeneration and oxidative stress in the brain, increase in DNA damage and oxidative stress in blood cells, and inflammation and damage to the liver and lungs [33,37,38,45,61]. However, in vitro experiments looking at the effects of long-term exposure are lacking in the AgNP toxicology literature. Most in vitro studies limit AgNP exposure to 24–72 h [25,32,37,38,39,40,41,42,43,44,45], with one report extending in vitro exposure to three weeks and reporting no toxicity from AgNP exposure to human corneal epithelial cells or murine eye-associated macrophages [62]. However, in this study, only one assay, which indirectly estimated cell death, was used to assess cytotoxicity from AgNP exposure. In our study, we performed treatments for up to six weeks, a time window that is much longer than any other in vitro study on AgNP toxicity. Moreover, by using a combination of microscopy-based analyses, we were able to show that AgNP exposure led to the accumulation of an increasing number of cells with abnormal DNA content, even at very low doses (Figure 7 and Figure 8). This was an important finding, given that growth curve analysis (Figure 4a, Figure 5a, and Figure 6a) did not reveal any major effect from AgNP exposure on the overall ability of cell populations to proliferate. Thus, our work indicates that cellular toxicology investigations should employ multiple assays, some of which should rely on single-cell analysis, to enable strong and comprehensive conclusions.

### 4.2. Cells with Abnormal DNA Content Emerge as a Result of AgNP-Induced Cell Division Defects

Our finding that exposure to AgNPs can induce micronuclei is consistent with previous studies, which reported micronucleus formation upon exposure to AgNPs in human lung epithelial cells and fibroblasts [39,52,63], hepatoma cells [45], and keratinocytes [53]. However, by combining micronucleus analysis with immunofluorescence for kinetochore proteins (Figure 7), we were able to show that micronuclei resulting from chromosome missegregation events (MN+) increased more substantially in response to AgNP exposure than micronuclei resulting from DNA damage (MN-), indicating that AgNP exposure can cause chromosome missegregation during mitosis. An unexpected observation was that the increase in micronuclei was not dose-dependent. This may depend on the fact that higher cell death rates were observed at higher doses. This cell death could deplete micronucleated cells to a greater extent than non-micronucleated cells due to hypersensitivity of micronucleated cells. Indeed, the increased cell death of micronucleated cells has been reported in other contexts [64]. Another major finding of our study was that exposure to AgNPs induced the accumulation of cells with two nuclei, as well as large and/or abnormally shaped nuclei (Figure 8a,b). These phenotypes indicate abnormal or incomplete cell division. In the present study, we did not investigate the exact cause of chromosome missegregation and/or abnormal cell division in cells exposed to AgNPs. Nevertheless, it is possible that the intracellular accumulation of AgNPs (Figure 2 and Figure 3) may lead to physical hindrance and interference with the proper function of the mitotic apparatus. Chromosome movement, assembly and function of the microtubule-based mitotic spindle, and/or assembly and function of the cytokinetic ring (responsible for cytoplasmic division at the end of mitosis) could all be potential targets of physical hindrance. For instance, asbestos, which is known to increase the risk of mesothelioma [65,66], has been referred to as a physical carcinogen, due to its ability to induce tumor initiation and progression, despite its inability to induce gene mutations [67,68]. Studies at the single-cell level provided evidence that asbestos fibers can physically interact with the mitotic chromosomes and interfere with their movement [69] and can sterically inhibit cytokinesis [70]. Importantly, the mechanisms of toxicity of carbon nanotubes and long silver nanowires are believed to share similarities with asbestos [71,72] and disruption of mitosis by single-walled carbon nanotubes was reported for relatively low levels of exposure [73]. Our findings show that AgNPs can interfere with cell division and the resulting abnormal cells can keep proliferating, suggesting that AgNP exposure could induce unpredictable and deleterious effects on human health.

### 4.3. The Health Risk Associated with Cell Division-Dependent Defects

The genetic changes arising from micronucleus formation or formation of cells with abnormal nuclear DNA content represent serious health risk factors. Indeed, micronuclei are commonly found in cancer cells [74] and high frequencies of micronuclei are a biomarker for increased risk of cancer [75]. It is likely that many of the micronuclei found in cancer cells contain whole chromosomes (positive for kinetochore proteins, MN+), given that anaphase lagging chromosomes are the most common chromosome segregation defect observed in cancer cells [76,77] and that lagging chromosomes can form micronuclei upon mitotic exit [78]. A number of studies have shown that DNA enclosed in micronuclei can accumulate damage [79,80,81]. In extreme cases, whole chromosomes enclosed in micronuclei can experience chromothripsis, a catastrophic chromosome rearrangement event that was initially identified in cancer cells [20,82]. Thus, our finding that prolonged AgNP exposure can increase the number of micronuclei (particularly those containing whole chromosomes; Figure 7) indicates that AgNPs can promote some of the genetic changes found in cancer cells. The emergence of cells with large, abnormally shaped nuclei (Figure 8a,b) was likely the result of cells exiting mitosis without proper chromosome or cell separation, which gives rise to daughter cells with double genome content. Importantly, ample evidence has shown that tetraploidy (double genome content) contributes to both tumor initiation and tumor progression. For instance, tetraploid, but not diploid, mammary epithelial cells were shown to induce subcutaneous tumors in nude mice [83]. Moreover, it was reported that nearly 40% of all tumors might have undergone whole genome duplication during clonal evolution of the primary tumor [84]. Finally, a recent study showed that tetraploid cells are enriched at the invasive front of primary tumors [85]. Thus, our finding that cells with increased DNA content emerge upon treatment provides another potential link between AgNP exposure and acquisition of phenotypes typically associated with cancer. Our observation that some of these cells can re-enter mitosis (Figure 8c) indicates that these abnormal mother cells can give rise to progeny with abnormal genomes, thus contributing to the accumulation of genome instability in the overall population.

## 5. Conclusions

In the present study, we investigated the effects of AgNP exposure on human, non-transformed epithelial cells. We aimed to increase the relevance of our findings by making unique choices in our experimental design. First, we chose to investigate a variety of treatment durations, including long-term exposure. Secondly, we used a range of concentrations, including low AgNP doses. Lastly, we performed assays that would allow us to gather information at the single-cell level and to specifically assess the outcomes of cell division errors. Overall, our work demonstrates that the choice of specific assays for cellular toxicology is crucial, given that only minimal effects may be apparent using certain approaches, but deleterious effects may be revealed when undertaking other types of analysis. Specifically, we showed that cell division-based, microscopy assays can provide an extra level of critical information compared to other traditional assays. Indeed, the approach used here revealed previously unseen cellular phenotypes arising upon long-term exposure to AgNPs. The cell-division based analysis presented here could, in the future, be used for in vitro toxicology studies using other cell types or more complex experimental systems, such as 3D tissue models.

## Figures and Tables

**Figure 1 ijerph-16-02061-f001:**
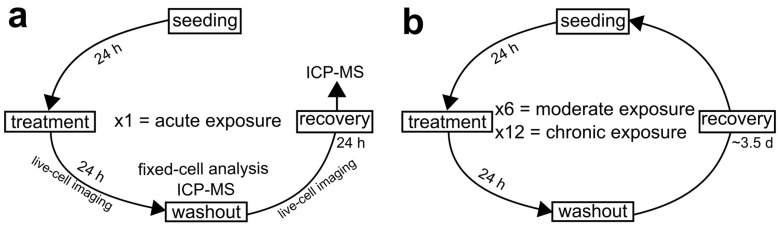
Diagrammatic representation of the experimental design conceived for the present study. (**a**) Experimental scheme for acute AgNP exposure where cells undergo one, 24 h AgNP treatment. (**b**) Experimental scheme for moderate and chronic AgNP exposure. For these prolonged exposures, acutely treated cell populations were maintained in culture and repeatedly exposed to AgNPs. After recovery, cells were passaged and exposed to another AgNP treatment. This cycle was repeated either six times (moderate exposure) over a period of three weeks or twelve times (chronic exposure) over a period of six weeks. In all experiments, live-cell imaging was performed in cells undergoing the last treatment cycle and during the 24 h following washout. Moreover, fixed-cell analysis was performed at the end of the last treatment cycle.

**Figure 2 ijerph-16-02061-f002:**
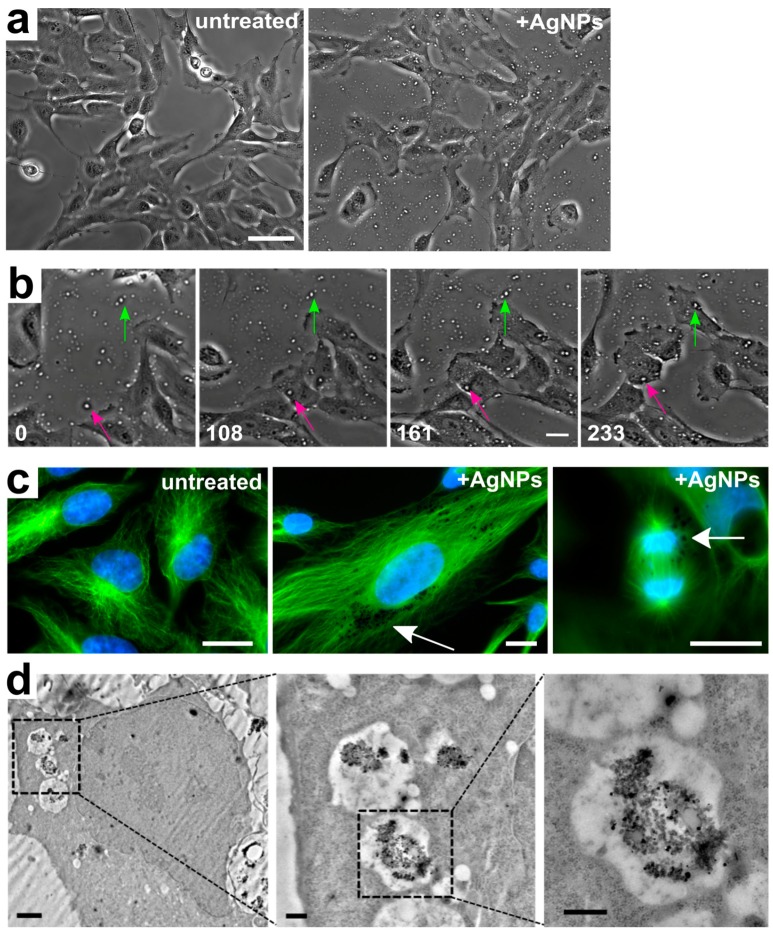
Silver nanoparticle internalization. (**a**) Still images from untreated (**left**) and AgNP-treated (**right**) live cell samples showing the presence of refractive dots (AgNPs) on and around the cells in the treated (**right**), but not in the untreated (**left**) sample (see also Appendix A). Scale bar, 10 μm. (**b**) Still images from time-lapse microscopy showing AgNPs becoming engulfed by individual cells. The green and pink arrows point at particles that are extracellular in the initial frame, but become engulfed by individual cells at subsequent time points and persist as intracellular particles (see also Appendix A). Numbers indicate elapsed time in min; Scale bar, 10 μm. (**c**) Representative images of fixed cells fluorescently stained for microtubules (**green**) and DNA (**blue**). The left image shows untreated cells. The middle and right images show an interphase and a mitotic cell, respectively, treated with 50 µg/mL of AgNPs. Arrows point to dark fluorescence-exclusion regions, assumed to be pockets of intracellular AgNPs. Scale bars, 10 μm. (**d**) Progressively magnified images from a transmission electron micrograph showing an example of a cell with internalized AgNPs contained in intracellular vesicles. Scale bars, 2 µm (**left panel**), 500 nm (**middle** and **right panels**).

**Figure 3 ijerph-16-02061-f003:**
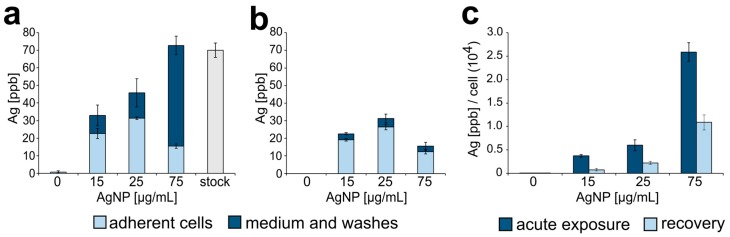
Silver nanoparticle intracellular accumulation. (**a**) Concentration of silver (ppb) found in adherent cells (light blue portion of bars) or in the extracellular medium (dark blue portion of bars) following acute exposure to AgNPs. The far-right grey bar shows the concentration of silver in a stock suspension of AgNPs prepared at a concentration of 75 µg/mL. (**b**) Concentration of silver (ppb) found in adherent cells (light blue portion of bars) or in the extracellular medium (dark blue **portion of** bars) following a 24 h recovery period from acute exposure. (**c**) Silver concentration normalized to the number of adherent cells in each sample.

**Figure 4 ijerph-16-02061-f004:**
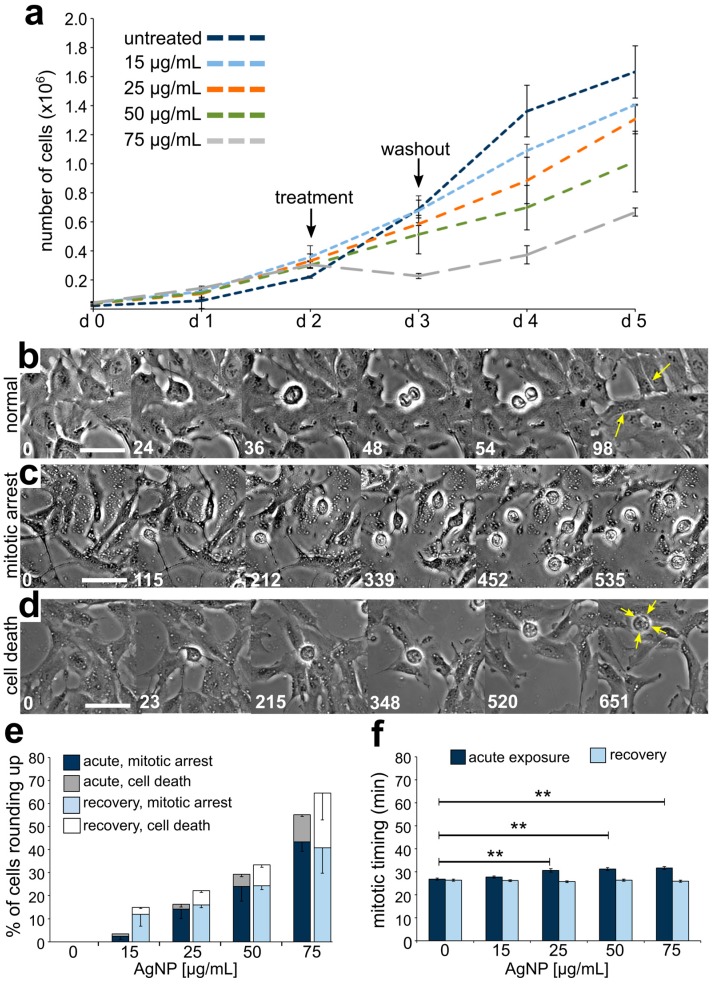
Effects on cell division and cell proliferation from acute AgNP exposure and recovery. (**a**) Growth curves for cells treated with increasing concentrations of AgNPs. (**b**–**d**) Still frame images from time-lapse microscopy showing: (**b**) a cell undergoing normal mitosis (yellow arrows pointing at the two resulting daughter cells), (**c**) cells rounding up and remaining rounded for more than 3 h (considered arrested), and (**d**) a cell rounding up and subsequently displaying signs of cell death (yellow arrows). Numbers indicate the elapsed time in min; Scale bars, 10 μm. (**e**) Percent of cells observed to either arrest or die after rounding up. The data corresponding to events observed during treatment are shown in darker colors (dark blue and grey) whereas the data corresponding to events observed during recovery are shown in lighter colors (light blue and white). (**f**) Mitotic timing in cells completing mitosis during an acute AgNP treatment (dark blue bars) or recovering from an acute AgNP treatment (light blue bars). ** t-test, *p* < 0.005

**Figure 5 ijerph-16-02061-f005:**
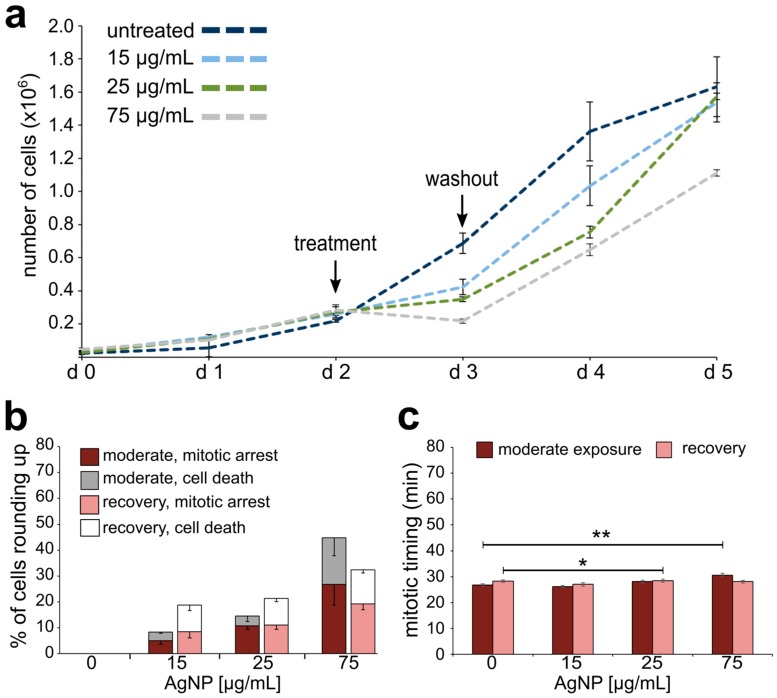
Effects on cell division and cell proliferation from moderate AgNP exposure and recovery. (**a**) Growth curves for cells exposed to a moderate treatment regime with increasing concentrations of AgNPs. (**b**) Percentage of cells observed to either arrest or die after rounding up. The data corresponding to events observed during the last of six treatments are shown in darker colors (**maroon** and **grey**), whereas the data corresponding to events observed during recovery from the last treatment are shown in lighter colors (**pink** and **white**). (**c**) Mitotic timing in cells completing mitosis during the last treatment of a moderate AgNP exposure (**maroon bars**) or recovering from the last treatment of a moderate AgNP exposure (**pink bars**). * t-test, *p* < 0.05; ** t-test, *p* < 0.005.

**Figure 6 ijerph-16-02061-f006:**
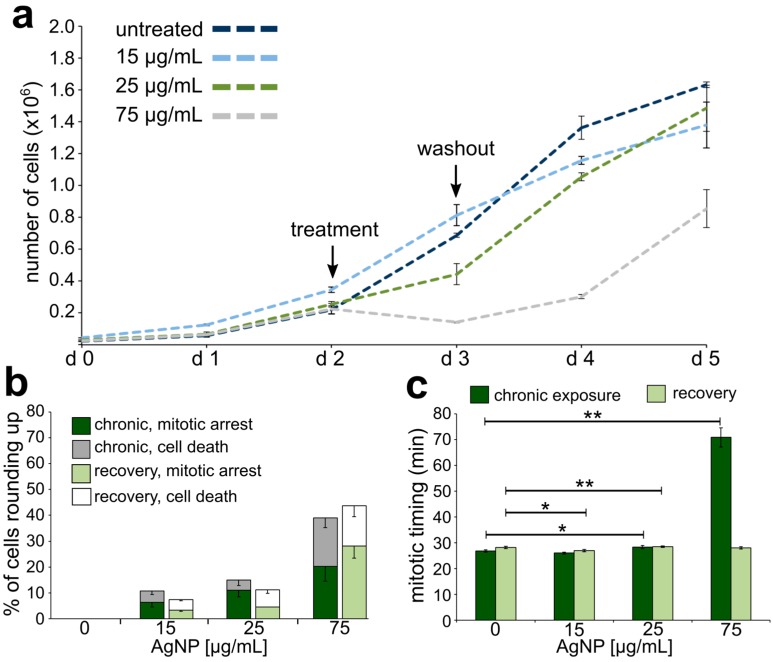
Effects on cell division and cell proliferation from chronic AgNP exposure and recovery. (**a**) Proliferation rates for cells exposed to a chronic treatment regime with increasing concentrations of AgNPs. (**b**) Percentage of cells observed to either arrest or die. The data corresponding to events observed during the last of 12 treatments are shown in darker colors (dark green and grey), whereas the data corresponding to events observed during recovery from the last treatment are shown in lighter colors (light green and white). (**c**) Mitotic timing in cells completing mitosis during the last treatment of a chronic AgNP exposure (dark green bars) or recovering from the last treatment of a chronic AgNP exposure (light **green** bars). * t-test, *p* < 0.05, ** t-test, *p* < 0.005.

**Figure 7 ijerph-16-02061-f007:**
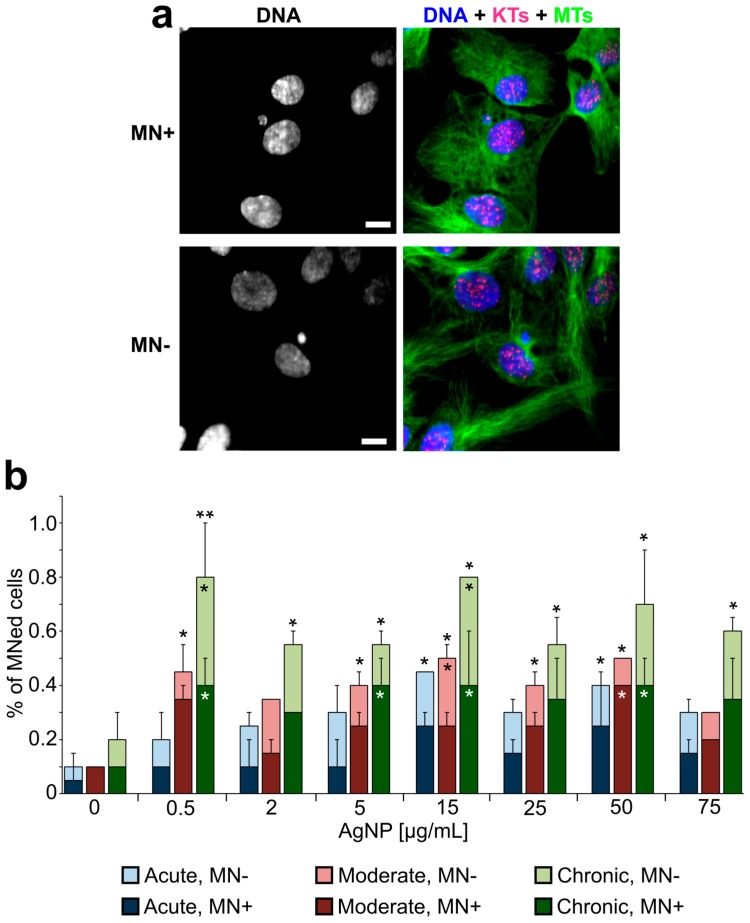
Increased frequencies of micronuclei in AgNP-treated cell populations. (**a**) Example fluorescence images with DNA staining (shown alone in the left panel and in blue in the right panel) and immunostaining for microtubules (MTs, green in the right panel) and kinetochore proteins (KTs, red/pink in the right panel). The two images represent examples of a kinetochore positive micronucleus (MN+, top; note red/pink kinetochore dot within the micronucleus) and a kinetochore negative micronucleus (MN-, bottom). Scale bars, 5 μm. (**b**) Percentage of cells with micronuclei (MN+ dark bars, MN- light bars) at the end of acute, moderate, or chronic exposure to AgNPs. *Fisher’s exact test, *p* < 0.05; **Fisher’s exact test, *p* < 0.005. The asterisks above the bars refer to comparisons for total number of micronucleated cells in treated samples vs. corresponding untreated control. The asterisks overlaid to individual bars refer to comparisons for a specific class of micronuclei (MN+, dark bars; MN- light bars) in treated samples vs. corresponding untreated control.

**Figure 8 ijerph-16-02061-f008:**
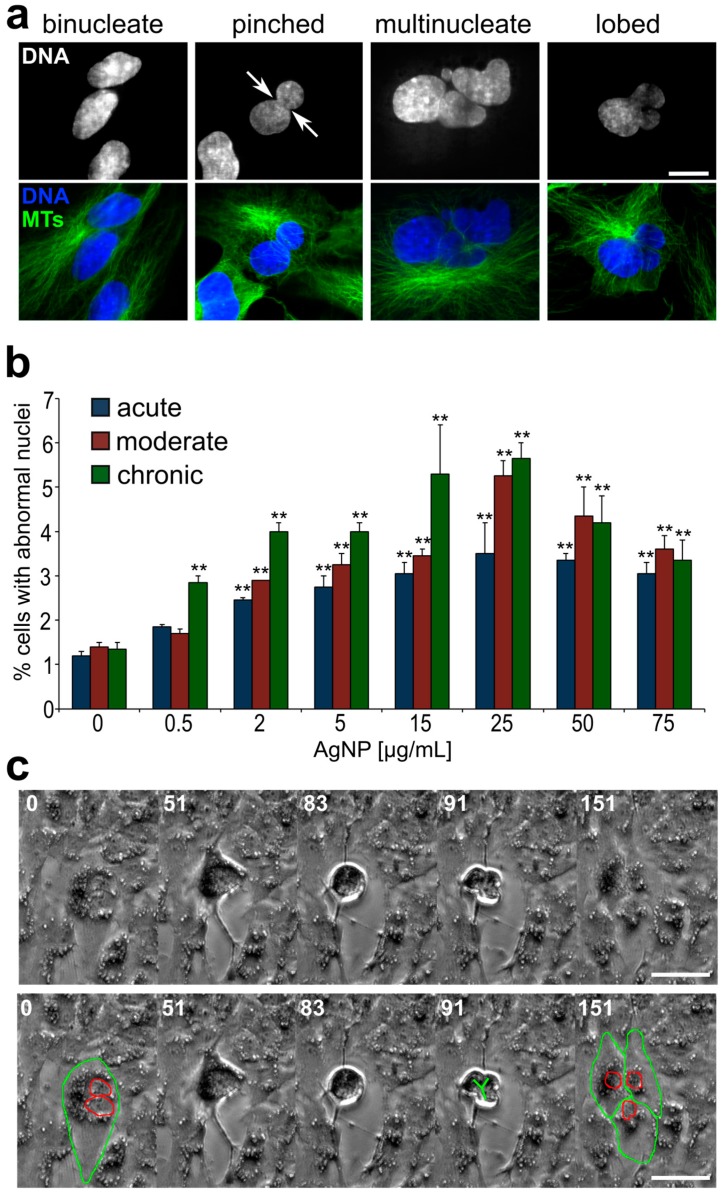
Increased frequencies of cells with large, abnormally shaped nuclei in AgNP-treated cell populations. (**a**) Example images of abnormal nuclear phenotypes quantified in this study. Images in the top row display DNA staining, whereas an overlay of DNA (blue) and microtubule (MTs, **green**) staining is shown in the bottom row. Scale bar, 10 μm. (**b**) Frequencies of cells with abnormal nuclear phenotypes following acute, moderate, or chronic AgNP exposure. **Fisher’s exact test, *p* < 0.005. (**c**) Still frames from time-lapse microscopy video recorded during the last treatment of a moderate exposure regime and showing a binucleate cell (highlighted in bottom panel) undergoing tripolar division (see also Appendix A), resulting in three daughter cells (highlighted in bottom panel). Numbers indicate elapsed time in min; Scale bar, 10 μm.

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
