# Peer review of "Single-Cell Analysis Reveals that Chronic Silver Nanoparticle Exposure Induces Cell Division Defects in Human Epithelial Cells"

_ijerph, 2019, doi:10.3390/ijerph16112061_

Round 1
Reviewer 1 Report
The manuscript by Cimini and collaborators deals with toxicity of chemical silver nanoparticles (AgNPs) and presents in vitro cell test. In general, it is nicely written, but needs some adjustments – such as in AgNP characterizations – for example, in material and methods section, it was stated that AgNPs had 20 nm in diameter – and, suddenly, in the manuscript appeared as triple or four-times in size (up to 95 nm). Also, the authors did not justify or explain why did they use such as great quantities of AgNPs – as none of the articles reported on activity of silver nanoparticles (antibacterial of antifungal) above of 20 μg/mL, which is also considered as “not satisfactory” activity – the best antimicrobial ativities are as low as 0.5-2.5 μg/mL and those “common doses” are significantly lower to ones that were used in the presented research. Other problem that was not discussed is with a model used to check the toxic effects of AgNPs in vitro– if intention was to substitute testing in animals – why the authors used cell culture – why did not they use tissue model – such as artificial skin for example? Silver nanoparticles are mostly used in topical applications, therefore, they would distribute in different manner in skin when compared to cell culture – where the exposure is direct. Finally, the last comment that needs attention is referencing, which needs an update.
Some minor problems are:
1) Use SI units – it is recommended to use mol/L, h, min, μL/mL etc.;
2) Avoid mixing the text that belongs to materials with discussion and/or results sections;
3) Better referencing – many recently published articles were omitted – there are no references published along 2017 and 2018;
4) Better legends for Figures – for example, Figure 2 – numbers and bars for magnification for (d) right, middle and left panels – need to be observable – 500 nm was cited twice, and looks like the right image was taken with 50 nm?;
5) Many figures do not have complete legends – some numbers that appear in Figures are not cited in Legends;
6) Authors do not tell about AgNPs destination, interaction with membrane, proteins their half-time life? How stable where the purchased particles?
Author Response
Reviewer 1
Comments and Suggestions for Authors
The manuscript by Cimini and collaborators deals with toxicity of chemical silver nanoparticles (AgNPs) and presents in vitro cell test. In general, it is nicely written, but needs some adjustments – such as in AgNP characterizations – for example, in material and methods section, it was stated that AgNPs had 20 nm in diameter – and, suddenly, in the manuscript appeared as triple or four-times in size (up to 95 nm). Also, the authors did not justify or explain why did they use such as great quantities of AgNPs – as none of the articles reported on activity of silver nanoparticles (antibacterial of antifungal) above of 20 μg/mL, which is also considered as “not satisfactory” activity – the best antimicrobial ativities are as low as 0.5-2.5 μg/mL and those “common doses” are significantly lower to ones that were used in the presented research. Other problem that was not discussed is with a model used to check the toxic effects of AgNPs in vitro– if intention was to substitute testing in animals – why the authors used cell culture – why did not they use tissue model – such as artificial skin for example? Silver nanoparticles are mostly used in topical applications, therefore, they would distribute in different manner in skin when compared to cell culture – where the exposure is direct. Finally, the last comment that needs attention is referencing, which needs an update.
The reviewer raises three major issues in the statement above. Our responses are as follows:
1. The first issue concerns the size of the AgNPs. As we indicate in the manuscript (page 3, lines 102-105), the AgNPs used in our study were obtained from a commercial source and the reported size is based on the company’s reported characterization. Our own characterization was performed on NPs suspended in water and we believe that the larger size depends on some degree of aggregation of the AgNPs. To address this point and major point 1 by reviewer 2, we have now also characterized the AgNPs suspended in culture medium. As now shown in Figure S1b, when the NPs are suspended in culture medium at concentrations within our experimental range, the size is substantially smaller, although still larger than 20 nm. Aggregation in biological fluids has been previously reported in the literature (Krystek et al., 2015 and Maiorano et al., 2010). The text has been revised (page 3, lines 102-116 and page 6, lines 265-282) to describe the new data and better explain where all the different sizes come from.
2. The second issue concerns the range of doses used in our study. Although we realize that some studies have used lower doses than ours, the range of doses used in our experiments falls within the range of many other toxicology reports in the literature. We have added references in the manuscript and slightly revised the text (page 3, lines 129-130 and references therein) to point this out.
3. The third question raised by the reviewer is why we used cells in culture for our study. We would like to point out that our main goal here was to develop a new cell division-focused approach to be used in toxicology studies. We realize, however, that the way our introduction was laid out did not make this point clearly. To clarify this, we added some text in the introduction, including a final paragraph (page 2, lines 72-73 and lines 88-95) while removing a paragraph that was in the earlier version and that took the focus away from this main goal. In the added text, we explain that RPE-1 cells are considered the gold standard to study cell division in non-transformed cells. We also point out that it is important to use such a standard experimental model of cell division as a first step toward optimizing cell division-based toxicology approaches, with the goal to later apply these approaches to other experimental models, including primary cell lines and whole-tissue models. We re-iterated this last point also in the conclusions section (page 17, lines 603-605). We hope that these changes will make the rationale behind our choice apparent.
Some minor problems are:
1) Use SI units – it is recommended to use mol/L, h, min, μL/mL etc.;
We have revised both text and figures according to the reviewer’s suggestion.
2) Avoid mixing the text that belongs to materials with discussion and/or results sections;
In response to this point and to a point raised by reviewer 3, we moved the paragraph that described our experimental design along with the corresponding figure (Figure 1) to the methods section (bottom of page 3 – top of page 4).
3) Better referencing – many recently published articles were omitted – there are no references published along 2017 and 2018;
We have added numerous references, including a number from the past two years.
4) Better legends for Figures – for example, Figure 2 – numbers and bars for magnification for (d) right, middle and left panels – need to be observable – 500 nm was cited twice, and looks like the right image was taken with 50 nm?;
We are not sure what the reviewer means by “observable.” In the version of the manuscript we submitted, the scale bars for Figure 2d were displayed (as thick black lines) at the bottom left corner of each image. The scale bars for the middle and right panels both represented 500 nm. However, the scale bar for the right panel was longer. We slightly changed the figure legend (last line) for clarity. Other changes were also made to the figure legends in response to other reviewers’ comments.
5) Many figures do not have complete legends – some numbers that appear in Figures are not cited in Legends;
All the figures had complete legends. However, some of the legends continued at the top of the following page. So, it is possible the reviewer missed them. It is not clear to us what the reviewer means by “some numbers that appear in Figures are not cited in Legends.”
6) Authors do not tell about AgNPs destination, interaction with membrane, proteins their half-time life? How stable where the purchased particles?
Some of the questions raised above appear to be beyond the scope of our study. But one thing we showed, in several ways, was that the AgNPs entered the cell rather than remaining associated with the plasma membrane, as NP internalization would likely be essential for the NPs to interfere with cell division (which was the focus of our study). We do not know explicitly how stable the purchased NPs are. However, we never kept the NPs for very long as each batch came with an expiration date of six months and therefore a new batch would be purchased. This is mentioned in the Materials and Methods section (page 3, lines 116-118) and is the reason why in Figure S1a we display DLS data from different batches.
Reviewer 2 Report
This study investigated the toxicity of in vitro exposure to different concentrations of silver nanoparticles (AgNP) for different exposure durations using the human hTERT-immortalized retinal pigmented epithelial (RPE-1) cell line. Multiple in vitro endpoints were investigated, including cellular uptake, cell viability, live cell imaging and analysis. The major findings were: (1) the selection of specific assays for cellular toxicity study is crucial, because only minimal effects may be apparent using certain approaches, but toxic effects may be revealed when undertaking other types of cellular analyses; (2) cell division-based, microscopy assays can provide an extra level of critical information compared to other traditional assays. Overall, the study was properly conducted and the manuscript was well written. However, this reviewer has some comments that may improve the manuscript. Major and minor points are listed below.
Major points:
1. Lines 93-104: In terms of characterization of the physicochemical properties of the nanoparticles, the authors only measured the hydrodynamic diameter, which is not sufficient. Nowadays, for any toxicity studies of nanoparticles, it is essential to fully characterize the nanoparticles, including the TEM size, hydrodynamic size, Zeta potential, shape, polydispersity index, etc. The authors measured the hydrodynamic size of nanoparticles in water. This is good, but the authors should also measure the hydrodynamic size of the nanoparticles in the cell culture media. The authors should fully characterize their nanoparticles and report the results.
2. In the literature, there are a lot of toxicity studies of silver nanoparticles. It is well known that silver nanoparticles can release the silver ion in liquid solution and the silver ion is more toxic than the silver nanoparticles. In this study, only silver nanoparticles are used. It is unknown the extent of dissolution of the silver nanoparticles in the cell culture media. Was the observed toxicity due to silver nanoparticles or due to the silver ion? This reviewer suggests that the authors include a group of silver ion (e.g., silver acetate) to elucidate this point.
3. In this study, an immortalized cell line is used. However, immortalized cells are different from the primary human cells. So the results may not be the same if using primary human cells. If the use of primary human cells is not possible, at least this point should be discussed in the manuscript.
4. Lines 119-130: Three exposure paradigms were employed to mimic acute, moderate, and chronic AgNP exposure. Were these exposure paradigms new and designed by the authors and used for the first time in this study? How did these exposure paradigms correspond to acute, moderate, and chronic in vitro exposures, respectively? Could the authors justify the rationale of these designs?
5. Lines 119-130: Exposure durations were described, but what about the exposure doses? Were the in vitro exposure doses relevant to realistic human exposures?
6. Lines 316-318: The cellular uptake results should be interpreted carefully. First of all, it was concluded that “the amount of silver taken up by individual cells is proportional to the amount of AgNPs available”. This may be valid only when the amount of AgNP has not reached the maximum uptake capacity of the cells. If using a higher concentration of AgNP, this statement may not be true anymore. Also, it is said that “once cells uptake the AgNPs, they will not expel them”. Based on the available limited result, I do not think the authors can draw this conclusion. In the field of nanotoxicology, it is well known that most cell types can release nanoparticles following uptake. This process is termed exocytosis. This rate of this process may be slow depending on the cell type, the type of nanoparticles, and the amount of nanoparticles. However, the authors should not conclude that cells do not expel AgNP based on their limited results.
Minor points:
7. Line 99: it is indicated that 12 different suspensions from 3 different batches, but in the Figure S1 legend, it is said that 12 individual suspensions from 4 different batches? So was it from 3 batches or 4 batches? Please clarify?
8. Line 96: did the authors use TEM to confirm the size is 20 nm. According to Figure S1, the hydrodynamic size is much bigger than the TEM size (70.15 nm vs. 20 nm).
9. Lines 122-124: There is an inconsistence between the text and Figure 1a. In the text, it is said that “For fixed-cell analysis of acutely treated cells, the cells were fixed at the end of the 24 hour treatment or after 24 hour of recovery in fresh media (Figure 1a)”. However, in Figure 1a, fixed-cell analysis is only labeled at the end of 24 h treatment. It should also be labeled after the 24 h recovery.
10. Lines 441-452: the number of cells with micronuclei did not appear to be dose-dependently increased? This is different from other endpoints that appear to be dose-dependent. Can the authors explain the possible reasons?
11. Lines 469-480: It is said that “Data are reported as mean ± SEM and represent the average of two independent experiments”, so what was the number of replicates for each bar in Figure 8b? Or what was the number of replicates in each experiment? This comment applies to all figures and all experiments. Please clarify this in the Methods and in all figure legends.
Author Response
Reviewer 2
This study investigated the toxicity of in vitro exposure to different concentrations of silver nanoparticles (AgNP) for different exposure durations using the human hTERT-immortalized retinal pigmented epithelial (RPE-1) cell line. Multiple in vitro endpoints were investigated, including cellular uptake, cell viability, live cell imaging and analysis. The major findings were: (1) the selection of specific assays for cellular toxicity study is crucial, because only minimal effects may be apparent using certain approaches, but toxic effects may be revealed when undertaking other types of cellular analyses; (2) cell division-based, microscopy assays can provide an extra level of critical information compared to other traditional assays. Overall, the study was properly conducted and the manuscript was well written. However, this reviewer has some comments that may improve the manuscript. Major and minor points are listed below.
Major points:
1. Lines 93-104: In terms of characterization of the physicochemical properties of the nanoparticles, the authors only measured the hydrodynamic diameter, which is not sufficient. Nowadays, for any toxicity studies of nanoparticles, it is essential to fully characterize the nanoparticles, including the TEM size, hydrodynamic size, Zeta potential, shape, polydispersity index, etc. The authors measured the hydrodynamic size of nanoparticles in water. This is good, but the authors should also measure the hydrodynamic size of the nanoparticles in the cell culture media. The authors should fully characterize their nanoparticles and report the results.
As indicated in the Materials and Methods section of the manuscript (page 3, lines 102-105 of revised manuscript), the NPs used in our study were obtained from a commercial source and characterized at the origin. As far as we can tell, studies (including publications from the past two years) in which particles are synthesized in-house include a full characterization, whereas toxicology studies that use commercial NPs do not provide a full characterization, as this is typically performed by the manufacturer. However, in this revised version of the manuscript we have added additional information (polydispersity index and zeta potential) for AgNPs suspended in water. Moreover, information is now also provided for AgNPs suspended in tissue culture medium. This information has been included in Figure S1 and described in the text (page 3, lines 106-116 for description of methods and page 6, lines 272-282 for description of results).
2. In the literature, there are a lot of toxicity studies of silver nanoparticles. It is well known that silver nanoparticles can release the silver ion in liquid solution and the silver ion is more toxic than the silver nanoparticles. In this study, only silver nanoparticles are used. It is unknown the extent of dissolution of the silver nanoparticles in the cell culture media. Was the observed toxicity due to silver nanoparticles or due to the silver ion? This reviewer suggests that the authors include a group of silver ion (e.g., silver acetate) to elucidate this point.
Based on this reviewer’s suggestion, we performed experiments aimed at determining the potential contribution of silver ions released from the AgNPs and the data are presented in an extensively revised version of Figure S3. First, we showed that cells treated with AgNO3, which is a source of silver ions, caused extensive cell death. However, when we added a silver ion chelator (NAC) along with AgNO3, cell death was suppressed (Figure S3a). We then showed that the percentages of live and dead cells in the adherent and non-adherent cell populations treated with AgNPs (Figure S3b-c) were not significantly different from the percentages of live and dead cells in the adherent and non-adherent cell populations treated with AgNPs along with NAC (Figure S3d-e), indicating that, within the AgNP dose range used in our experiments, there is no major effect due to release of silver ions. The data presented in Figure S3 are also discussed in the revised text (page 10, lines 389-405).
3. In this study, an immortalized cell line is used. However, immortalized cells are different from the primary human cells. So the results may not be the same if using primary human cells. If the use of primary human cells is not possible, at least this point should be discussed in the manuscript.
We realize that we did not explain our cell line choice well in the initial version of our manuscript. Our main goal here was to develop a new cell division-focused approach to be used in toxicology studies, but this point was also not made very clearly. To clarify this, we added some text in the introduction, including a final paragraph (page 2, lines 88-95) while removing a paragraph that was in the earlier version and that took the focus away from this main goal. In the added text, we explain that RPE-1 cells are considered the gold standard to study cell division in non-transformed cells. We also point out that it is important to use such a standard experimental model of cell division as a first step toward optimizing cell division-based toxicology approaches, with the goal to later apply these approaches to other experimental models, including primary cell lines and whole-tissue models. We re-iterated this point in the conclusions section (page 17, lines 603-605). We hope this addresses the reviewer’s concern.
4. Lines 119-130: Three exposure paradigms were employed to mimic acute, moderate, and chronic AgNP exposure. Were these exposure paradigms new and designed by the authors and used for the first time in this study? How did these exposure paradigms correspond to acute, moderate, and chronic in vitro exposures, respectively? Could the authors justify the rationale of these designs?
The experimental design was conceived by us. We added some text in the introduction (lines 95-96) and methods section (page 3, lines 131-139) to point this out. We also added references to indicate that the doses of AgNPs selected for our study were in the range of doses used in other studies (page 3, line 130). Furthermore, we revised the text to indicate that the terms “acute,” “moderate,” and “chronic” were used by us (page 3, line 132) to simply refer to a short, medium, and long treatment duration, as indicated in Figure 1. Finally, we added some text (page 3, line 135 and line 139) to explain the rationale behind our choices for treatment duration.
5. Lines 119-130: Exposure durations were described, but what about the exposure doses? Were the in vitro exposure doses relevant to realistic human exposures?
It is difficult to estimate what the realistic human exposure doses may be and these may differ depending on exposure source. However, as already noted in the response to the previous question, we have now clarified (and included relevant references; page 3, line 130) that the doses were within ranges used in many toxicology studies.
6. Lines 316-318: The cellular uptake results should be interpreted carefully. First of all, it was concluded that “the amount of silver taken up by individual cells is proportional to the amount of AgNPs available”. This may be valid only when the amount of AgNP has not reached the maximum uptake capacity of the cells. If using a higher concentration of AgNP, this statement may not be true anymore. Also, it is said that “once cells uptake the AgNPs, they will not expel them”. Based on the available limited result, I do not think the authors can draw this conclusion. In the field of nanotoxicology, it is well known that most cell types can release nanoparticles following uptake. This process is termed exocytosis. This rate of this process may be slow depending on the cell type, the type of nanoparticles, and the amount of nanoparticles. However, the authors should not conclude that cells do not expel AgNP based on their limited results.
We have toned down this conclusion by specifying that our conclusion is limited to the timeframe, dose range, and cell type used in our study (page 8, lines 341-344).
Minor points:
7. Line 99: it is indicated that 12 different suspensions from 3 different batches, but in the Figure S1 legend, it is said that 12 individual suspensions from 4 different batches? So was it from 3 batches or 4 batches? Please clarify?
We apologize for the confusion. Four different batches of nanopowder were analyzed and DLS measurements were taken for three suspensions per batch. This has been corrected in the manuscript (page 3, lines 114-116).
8. Line 96: did the authors use TEM to confirm the size is 20 nm. According to Figure S1, the hydrodynamic size is much bigger than the TEM size (70.15 nm vs. 20 nm).
TEM was used by the manufacturer to confirm the size of the nanopowder (mentioned on page 3, lines 103-105). AgNPs have been previously shown to aggregate to some extent in aqueous suspension and this also appears to be the case in our suspensions. Our new measurements of AgNPs suspended in tissue culture media (Figure S1b) indicate that there was minimal aggregation at concentrations within the range used in our experiments. The discussion of our new data can be found in the first paragraph of the results section (bottom of page 6).
9. Lines 122-124: There is an inconsistence between the text and Figure 1a. In the text, it is said that “For fixed-cell analysis of acutely treated cells, the cells were fixed at the end of the 24 hour treatment or after 24 hour of recovery in fresh media (Figure 1a)”. However, in Figure 1a, fixed-cell analysis is only labeled at the end of 24 h treatment. It should also be labeled after the 24 h recovery.
The error was in the text and it has been fixed.
10. Lines 441-452: the number of cells with micronuclei did not appear to be dose-dependently increased? This is different from other endpoints that appear to be dose-dependent. Can the authors explain the possible reasons?
This point has been added to the description of the results (page 13, lines 471-473) and discussed on pages 15-16, lines 543-547.
11. Lines 469-480: It is said that “Data are reported as mean ± SEM and represent the average of two independent experiments”, so what was the number of replicates for each bar in Figure 8b? Or what was the number of replicates in each experiment? This comment applies to all figures and all experiments. Please clarify this in the Methods and in all figure legends.
Information related to statistics has been removed from the legends and moved to individual sections of the Materials and Methods. In the figure legends, we only left information about statistical significance. It seemed to us that this was the best compromise to address this comment and the request of reviewer 3 to shorten the figure legends.
Reviewer 3 Report
The paper is well presented with well written discussion and conclusion.
The rationale of the study needs to be discussed in more details, i.e. why RPE-1 cells were chosen in this study.
The first section of the result and Figure 1 should be mentioned in the method section.
The figures’ legend can be summarized and some parts can be mentioned within the main text.
Author Response
Reviewer 3
The paper is well presented with well written discussion and conclusion.
We thank the reviewer for acknowledging the good quality of our manuscript.
The rationale of the study needs to be discussed in more details, i.e. why RPE-1 cells were chosen in this study.
As also pointed out in our responses to reviewer 1 (major point 3) and reviewer 2 (point 3), the goal of our study was to develop a cell division-based approach for toxicology studies and RPE-1 cells are currently considered the gold standard for studying cell division in non-transformed cells. Changes have been made to the introduction by eliminating one paragraph and adding a new paragraph that we believe makes the rationale of our study clearer (page 2, lines 88-95). We also added information in the Materials and Methods section aimed at better explaining the rationale of our experimental design (page 3, lines 131-139).
The first section of the result and Figure 1 should be mentioned in the method section.
Both the text and figure have been moved to the Materials and Methods section.
The figures’ legend can be summarized and some parts can be mentioned within the main text.
Some of the information related to experimental design and statistics has been removed from the legends and moved to individual sub-sections of the Materials and Methods. This has contributed to shortening most of the figure legends. Additional redundant information was also removed from the figure legends.
Round 2
Reviewer 2 Report
The authors have done a nice job addressing the majority of my comments adequately. I only have one question regarding my last comment on the number of replicates or the number of samples each experiment. The authors have tried to address this by adding information to the end of each related section of the Materials and Methods. However, based on the newly added information, I am still confused. For example, it is indicated that “Counts were obtained from two technical replicates for each sample and data are reported as the mean ± SEM from two independent experiments.” (lines 194-196). So from two independent experiments, did you get the mean value from these two experiments? How many samples in each experiment? I believe you got the average for each individual sample from two technical replicates as indicated in lines 174-176. Please indicate so in other places as well. These comments apply to other sections of the relevant methods. Basically, when I look at the figures, I just want to be clear how many replicates or how many samples were used to represent each bar in the figure. This should be clarified.
Also, in some places, the symbol “±” is used, whereas in other places, the symbol of “-” or “+” is used (lines 217, 235, and 263). Please correct this.